# Higher Diet Quality in Latina Women during Pregnancy May Be Associated with Sociodemographic Factors

**DOI:** 10.3390/ijerph192113895

**Published:** 2022-10-26

**Authors:** Juliana Teruel Camargo, Matthew K. Taylor, Byron J. Gajewski, Susan E. Carlson, Debra K. Sullivan, Heather D. Gibbs

**Affiliations:** 1Department of Dietetics & Nutrition, School of Health Professions, University of Kansas Medical Center, Kansas City, KS 66160, USA; 2Department of Urology, School of Medicine, University of Kansas Medical Center, Kansas City, KS 66160, USA; 3Department of Biostatistics & Data Science, School of Medicine, University of Kansas Medical Center, Kansas City, KS 66160, USA

**Keywords:** nutrition literacy, acculturation, diet intake, diet quality, pregnancy, Hispanic or Latino, immigrants

## Abstract

Acculturation contributes to low diet quality and can foster health inequities for Latina women during pregnancy. Conversely, nutrition literacy (NL) increases diet quality and could promote health equity. This study assessed the associations between the diet quality, acculturation, and NL of Latina women (n = 99) participating in the Assessment of Docosahexaenoic Acid On Reducing Early Preterm Birth (ADORE) study. Acculturation and nutrition literacy factored together tended to modify diet quality, but this was not statistically significant. Diet quality was associated with acculturation, age, and nativity. Most (76.8%) demonstrated low nutrition literacy. Women who were bicultural and were born in Latin American countries other than Mexico had lower diet quality scores than women who had lower acculturation and were born in Mexico. Women who were 35 years or older had better diet quality than those who were younger. Future studies are needed to explore diet quality differences for pregnant Latina women with high nutrition literacy and high acculturation, as well as for women from the Caribbean, Central and South American countries living in the US, to promote nutrition and maternal health for Latina women.

## 1. Introduction

A healthy diet is important for promoting a healthy pregnancy. Suboptimal nutrition during pregnancy can result in adverse fetal outcomes, and treatment guidelines for chronic diseases that first emerge during pregnancy include diet and exercise as primary approaches [1]. Unfortunately, many pregnant women in the US under-consume some essential nutrients (e.g., vitamins A, C, D, E, K, iron, calcium, folate, and others) while exceeding the recommended consumption of other nutrients (e.g., sodium) [2]. 

Health inequities exist among pregnant women living in the US. Women who are medically underserved experience a higher onset of nutrition-related acute and chronic diseases during pregnancy [3,4,5]. Among the potential factors contributing to nutrition-related diseases experienced by Latina women during pregnancy are nutrition literacy (NL) and acculturation. NL is “*the capacity to obtain, process, and understand nutrition information and skills needed to make appropriate nutrition decisions*” and is directly associated with the socioeconomic position of pregnant Latina women, where those with a lower socioeconomic position have lower nutrition literacy [6]. Acculturation is “*the process of learning and incorporating the attitudes, values, customs, beliefs, and behaviors of the mainstream culture of the new country immigrants and their families are living in*” and is inversely associated with the diet quality of pregnant Latina women [7]. While it is known that NL and acculturation as single factors have opposite effects on diet quality, the direction of their combined effect is unknown. The process of assimilating or adapting to a highly processed diet, aka, the Western diet, is a phenomenon observed among the Latino population in the US [8], but also in populations across Latin America [9,10]. However, the underrepresentation of Latina immigrant women in clinical and population studies leaves a problematic gap in the understanding of the factors contributing to the diet changes of pregnant Latina immigrants living in the US. Thus, using available data to explore potential factors that prevent or reverse the negative impact of diet acculturation has the potential to promote health equity for Latinos in the US and Latin America. Therefore, this study aimed to explore the potential differences between NL and acculturation to uncover potential trends in the diet quality of Latina women during pregnancy.

## 2. Materials and Methods

This cross-sectional study was nested within the phase III randomized clinical trial Assessment of Docosahexaenoic acid (DHA) to Reduce Early preterm birth (ADORE) and included women who self-identified as Hispanic or Latina from December 2016 to December 2018. Details of the methods and primary outcomes of the ADORE study are published elsewhere [11].

Participants were eligible for the nested study if they were ≥18 years, between 12 and 20 gestational weeks, had an acknowledged ability to speak and read either English or Spanish and were expecting a singleton. Ineligibility criteria included unwillingness to consume DHA capsules from the parent randomized clinical trial. The University of Kansas Medical Center, Kansas City, Institutional Review Board approved all study procedures. Informed consent was required of all study participants in their language of preference.

Diet and supplement intake were collected on 3 occasions, between 12 and 28 gestational weeks, using standardized quantitative 24-h dietary recalls [12]. Using Nutrition Data System for Research software (version 2017), food recalls were transformed into the nutrient intake and Healthy Eating Index 2015 (HEI-2015) score. The HEI-2015 is a validated method for quantifying diet quality, described by the US Dietary Guidelines. The total HEI-2015 score, ranging from 0 (poor) to 100 (high), was derived by summing 13 component scores [13]. Nutrition literacy (NL) was measured using the 42-item Nutrition Literacy Assessment Instrument, with scores interpreted as ≤28 “low NL”, 29–38 “moderate NL”, and ≥39 “high NL” [14,15]. Acculturation was measured with the general acculturation index, ranging from 0 to 5. The general acculturation index scores were interpreted as <2.40 “low acculturation”, 2.40–3.69 as “bicultural”, and ≥3.70 as “high acculturation” [16]. Participants self-reported age, education level, annual household income, nativity, and time living in the US. Age was categorized by fertility ranges (25 to 34 years and 35 and over years). Education was categorized as “less than high school degree” and “equal or more than high school degree graduate”. Annual household income was reported in ranges (less than USD 24,999 and USD 25,000 and over). Nativity was categorized as “USA”, “Mexico”, and Other (which included participants born in Caribbean, Central, and South America countries). Considering the previous literature showing that first-generation immigrants undergo diet changes when moving to the US mainland [16], nativity was categorized as ‘non-immigrant’ for participants born in the US mainland and ‘immigrant’ for participants born in the US territories and foreign countries. Time living in the US was only asked for immigrants and categorized in years as “less than 10 years” and “11 and over”.

Data were evaluated using descriptive statistics. All variables were tested for normality (Shapiro–Wilk normality test, skewness, and Q–Q plot of residuals) and outliers (a modified z-score method) [17]. Participants who completed a single 24 h recall (n = 13) or did not answer more than nine questions on the nutrition literacy (NL) tool [14] (n = 5) were excluded. All participants had plausible reported energy intake. After excluding participants with single 24 h recalls or with uncompleted NL questions and combining participants who had bicultural and high acculturation scores, the sample was normally distributed. General linear models were tested to explore differences in the diet quality of Latina women during pregnancy. Six models with diet quality (Healthy Eating Index-2015 (HEI-2015)) as the dependent variable were tested for relative model-fit with the following independent variables: (1) acculturation (bicultural/low acculturation); (2) NL (moderate/low); (3) acculturation and NL combined into four groups (low acculturation/low NL, low acculturation/moderate NL, bicultural/low NL, and bicultural/moderate NL); (4) acculturation (bicultural/low acculturation) adjusted by sociodemographic characteristics (age less than 35 (yes/no), nativity (Mexico, US, and other), living in the US more than 10 years (yes/no), education level (high school degree (yes/no)); (5) nutrition literacy (low/moderate) adjusted by sociodemographic characteristics (age less than 35 (yes/no), nativity (Mexico, US, and other), living in the US more than 10 years (yes/no), education level (high school degree (yes/no)); (6) acculturation and NL combined into four groups (low acculturation/low NL, low acculturation/moderate NL, bicultural/low NL, and bicultural/moderate NL) adjusted by sociodemographic characteristics (age less than 35 (yes/no), nativity (Mexico, US, and other), living in the US more than 10 years (yes/no), education level (High school degree (yes/no). We also conducted a sensitivity Analysis of Variance (ANOVA) model with acculturation and NL as continuous variables. The model selection was based on the Akaike information criterion (AIC), with the lowest AIC, and was also checked for homoscedasticity. The Tukey’s Honestly Significant Difference (Tukey’s HSD) was performed as a post hoc test to explore differences in diet quality. A p-value of 0.05 was considered statistically significant. Statistical analyses were performed using the R Foundation for Statistical Computing (Vienna, Austria) [18].

## 3. Results

In this sample of 99 Latina women during pregnancy, most were less than 35 years old, reported less than High school education, had less than USD 25,000 annual household income, immigrated to the US mainland, were born in Mexico, and lived in the US for 11 or more years (Table 1).

Participants achieved an average NL score of 24.3 out of 42 on the Nutrition Literacy Assessment Instrument; 76.8% had scores indicating low nutrition literacy (NL). For this sample, there were no high NL scores. The average acculturation score was 1.8 out of 5.0, and most (74.7%) had low acculturation, with only 4% having high acculturation. On average, participants achieved a diet quality score of 69.8. 

We found that model 4 was the best model (lowest Akaike information criterion (AIC) among the six models) with statistically significant difference in average diet quality (Healthy Eating Index-2015 (HEI-2015)) for acculturation (f(2) = 4.65, *p* = 0.03), age (f(2) = 5.27, *p* = 0.02), nativity (f(2) = 3.92, *p* = 0.02) and time living in the US (f(2) = 6.06, *p* < 0.02). Education level was part of the model but was not statistically significant (Table 2).

Diet quality scores varied by acculturation level, where participants who were bicultural reported an overall diet quality score 3.7 times lower than those who had low acculturation. Participants who were bicultural reported a lower score for all dietary components, except for dairy and saturated fats (Table 3). Diet quality score differences by nutrition literacy and acculturation level factored together can be found in the Appendix A.

## 4. Discussion

The findings of this study indicate that acculturation, age, and nativity were associated with the diet quality in this sample of Latina women during pregnancy. Participants who were older had better diet quality than those who were younger, while those who were born in a Latin American country other than Mexico or were classified as bicultural had lower diet quality than participants who were born in Mexico or were classified with low acculturation. 

Similar to previous studies, diet quality was positively associated with age [7]. A well-known generational difference, younger generations consistently report lower consumption of fruits, vegetables, and whole grains, while reporting a higher consumption of ultra-processed foods, such as foods higher in added sugars [8,9].

Consistent with other research, low acculturation appeared to be a potential protective factor for higher diet quality, where women with low acculturation consumed a healthier diet during pregnancy [19]. In fact, participants achieved a diet quality score of 69.8, which is a higher score than the average diet of Americans (58 out of 100) [20]. Acculturation is a complex phenomenon that can have both positive and negative impacts on the lifestyle of immigrants and their families. For example, high acculturation is associated with higher physical activity levels but a lower intake of fruits and vegetables among Latino populations [10]. How immigrants assimilate to the new country is impacted by multiple individual, interpersonal and systemic factors, such as their immigration history, previous interactions in their home country, and their interaction and integration in the new environment [10]. 

The highest diet quality was seen in this sample among those with low acculturation and low nutrition literacy, This suggests that, in settings where a traditional healthy diets are the social norm, nutrition literacy (NL) may be less critical for diet quality. As no participants in this study demonstrated a high NL, however, readers should exercise caution when drawing conclusions about the relationship between nutrition literacy and diet quality in pregnant Latina women. It remains unknown whether diet quality is different in the case of high NL and high acculturation. It is possible that, as women adapt to the new food environment where the traditional unprocessed and healthy diet is not accessible and promoted, and low-quality foods are widespread, NL becomes essential for selecting more healthful available food choices [21]. 

Interestingly, diet quality was significantly different among Latina women from Latin American countries other than Mexico compared to women born in Mexico, but was not different when compared with Latina women born in the US. In this way, our findings add to the literature that emphasizes the importance of considering the food experiences of Latin American immigrants in their home country, their history of immigration, and their experience navigating the new food environment when working with Latino communities that are underrepresented among the overall Latino community [10,22]. 

There are significant limitations to this study. The study sample is not generalizable to all Latino heritage groups representing the Latino population, as this sample consisted primarily of women of Mexican heritage. However, the representation is consistent for Latino subgroups in the Midwest area. Participants’ dietary intake and diet quality were self-reported, and there were no biomarkers to test the accuracy of self-reported food and meals. However, multiple 24 h dietary recalls are considered the gold standard for assessing individual dietary intake and the most appropriate method for culturally diverse and low-literacy populations [12]. 

## 5. Conclusions

In this study of Latina women during pregnancy, a majority of whom were immigrants and reported low education and income levels, diet quality was negatively associated with acculturation, nativity, time living in the US and age. These findings point to gaps in the literature that could lead to further studies that explore the diet quality and food choices of different nationalities within Latino communities. Further research should investigate whether diet quality during pregnancy differentiates among Latina women from the Caribbean, Central, and South American countries living in the US. Moreover, more studies are needed to understand the diet-quality differences for pregnant Latina women who have high acculturation levels and high nutrition literacy. These efforts could lead to interventions and policies that promote nutrition and maternal health equity for all pregnant Latina women. 

## Figures and Tables

**Table 1 ijerph-19-13895-t001:** Demographic characteristics of Latina women participating in a DHA supplement study during pregnancy.

Characteristics	Overall (n = 99)
	n (%)
Age, years	
Less or equal than 34	74 (74.7)
35 and over	25 (25.3)
Education level	
Less than high school degree	57 (57.6)
Equal or more than high school degree graduate	40 (42.4)
Annual household income, US dollars	
Less than $25,000	61 (61.6)
Equal or more than $25,000	38 (38.4)
Nativity	
Mexico	62 (62.6)
Other ^1^	24 (24.3)
US	13 (13.1)
Immigrants	
Yes	86 (86.9)
No	13 (13.1)
Time living in the US, years ^4^	
Equal or less than 10	38 (44.2)
11 and over	48 (55.8)

^1^ Argentina, Brazil, Dominican Republic, El Salvador, Guatemala, Honduras, and Venezuela; ^4^ only for those born outside of US mainland.

**Table 2 ijerph-19-13895-t002:** Average diet quality score differences by Analysis of Variance (ANOVA) tested models.

Characteristics	Comparison Groups	Model 1	Model 2	Model 3	Model 4	Model 5	Model 6
		HEI-2015 Mean Score Difference (95%CI)					
Acculturation	B:L	−5.18 (−10.31, −0.05) *			−5.18 (−9.97, −0.39) *		
Nutrition literacy	M:L		−1.50 (−6.73, 3.73)			−1.50 (−6.39, 3.39)	
Acculturation and nutrition literacy	BL:LL			−4.97 (−13.13, 3.18)			−4.97 (−12.58, 2.63)
	LM:LL			0.93 (−9.10, 7.23)			0.93 (−8.54, 6.68)
	BM:LL			−6.29 (−17.43, 4.86)			−6.29 (−16.69, 4.10)
	LM:BL			4.04 (−6.08, 14.17)			4.04 (−5.40, 13.49)
	BM:BL			1.32 (−13.97, 11.34)			1.31 (−13.12, 10.49)
	BM:LM			−5.36 (−18.02, 7.29)			−5.36 (−17.17, 6.44)
Age, years	35 and over: less than 35				5.58 (0.74, 10.42) *	5.87 (0.92, 10.84) *	5.57 (0.67, 10.47) *
Nativity	Mexico:US				5.10 (−2.23, 12.46)	7.77 (0.26, 15.29) *	5.10 (−2.32, 12.54)
	Other ^1^:US				−1.01 (−9.33, 7.31)	2.14 (−6.38, 10.67)	−1.09 (−9.53, 7.34)
	Other ^1^:Mexico				−6.11 (−12.07,−0.12) *	−5.63 (−11.74, 0.48)	−6.20 (−12.24, −0.16) *
Time living in the US, years	Equal or less than 10:11 and over				4.27 (−0.01, 8.56) *	3.54 (−0.85, 7.93)	4.26 (−0.08, 8.60) *
Education	Less than High school:High school or more				1.37 (−2.80, 5.55)	1.19 (−3.08, 5.45)	1.25 (−2.97, 5.47)
AIC		763.24	766.92	767.46	755.47	759.66	760.01

Abbreviations: CI, confidence interval; US, United States of America; HEI-2015, The Healthy Eating Index-2015; B, bicultural; L, low; M, moderate; BL, bicultural low nutrition literacy; LL, low acculturation and low nutrition literacy; LM, low acculturation and moderate nutrition literacy; BM, bicultural and moderate nutrition literacy; AIC, Akaike information criterion. Notes: The HEI-2015 is a measure of diet quality used to assess how well a set of foods aligns with the 2015–2020 Dietary Guidelines for Americans. The HEI-2015 includes 13 components that can be summed to a maximum total score of 100 points. A higher score indicates a diet that aligns with the Dietary Guidelines. * *p* < 0.05. ^1^ Argentina, Brazil, Dominican Republic, El Salvador, Guatemala, Honduras, and Venezuela.

**Table 3 ijerph-19-13895-t003:** Average diet quality score differences by acculturation.

Diet Quality Components	Maximum Points	Overall (n = 99)	Low Acculturation ^1^ (n = 83)	Bicultural ^1^ (n = 25)
		Mean Score (95%CI)		
Total HEI-2015	100	69.8 (67.8, 71.8)	69.6 (66.9, 72.3)	65.9 (59.9, 72.0)
Total fruits	5	3.5 (3.2, 3.8)	3.6 (3.2, 4.1)	3.4 (2.5, 4.4)
Whole fruits	5	3.9 (3.6, 4.2)	4.1 (3.7, 4.5)	3.8 (2.9, 4.8)
Total vegetables	5	3.7 (3.4, 4.0)	3.8 (3.4, 4.1)	3.5 (2.7, 4.3)
Greens and beans	5	3.6 (3.3, 4.0)	3.9 (3.4, 4.3)	3.4 (2.3, 4.5)
Whole grains	10	7.6 (6.9, 8.4)	6.9 (6.0, 7.7)	6.2 (4.3, 8.1)
Dairy	10	6.6 (6.1, 7.1)	6.6 (5.9, 7.3)	6.7 (5.1, 8.2)
Total protein foods	5	4.7 (4.5, 4.8)	4.7 (4.5, 4.9)	4.4 (4.0, 4.8)
Seafood and plant proteins	5	3.8 (3.5, 4.2)	3.9 (3.4, 4.4)	3.8 (2.7, 4.9)
Fatty acids	10	4.7 (4.1, 5.4)	4.2 (3.4, 5.1)	4.0 (2.1, 6.0)
Refined grains	10	7.5 (6.9, 8.1)	7.4 (6.6, 8.2)	6.6 (4.9, 8.4)
Sodium	10	4.6 (4.0, 5.3)	5.0 (4.2, 5.9)	4.5 (2.5, 6.4)
Added sugars	10	8.1 (7.6, 8.5)	8.0 (7.4, 8.6)	7.5 (6.1, 8.9)
Saturated fats	10	6.9 (6.3, 7.4)	6.9 (6.2, 7.6)	7.4 (5.8, 9.0)

Abbreviations: CI, confidence interval; HEI-2015, The Healthy Eating Index-2015; Notes: The HEI-2015 is a measure of diet quality used to assess how well a set of foods aligns with the 2015–2020 Dietary Guidelines for Americans. The HEI-2015 includes 13 components that can be summed to a maximum total score of 100 points. A higher score indicates a diet that aligns with the Dietary Guidelines. ^1^ Adjusted for age, nativity, time living in the US and education attainment.

## Data Availability

Not applicable.

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
