# Peer review of "Higher Diet Quality in Latina Women during Pregnancy May Be Associated with Sociodemographic Factors"

_ijerph, 2022, doi:10.3390/ijerph192113895_

Round 1
Reviewer 1 Report
While I appreciate the spirit of this study, the statistical analysis was not appropriate (testing for group differences with cross-sectional data without controlling for confounding variables and comparing groups of very small sample sizes). My specific comments are listed below.
Abstract line 16: Given your cross-sectional study design and statistical approach, I think describing it as “associations between” rather than “differences between” is more accurate.
Abstract line 17: Please report the sample size, e.g., “This study assessed differences between diet quality, acculturation, and NL of Latina women (n=XXX)…”
Abstract line 18: Please report the p value.
Line 32: Seems odd to exclude folate from your list of examples, given its primary role in fetal growth and development.
Line 68: It’s not clear what is meant by “which adds to a global score of 0 to 5.” The acculturation score gets added to what global score?
Lines 74-75: Annual household income should really be normalized by household size (i.e., an annual household income of 50k for a family of 5 is very different than 50k for a family of two. If you collected household size data, please normalize.
Table 1: Please summarize and test for differences in these characteristics by acculturation and NL groups. My suspicion is that there are statistically significant differences in these characteristics by group, which could be driving your results (see my next comment).
Lines 103-112: Given your cross-sectional survey design, just testing for statistical difference between groups without controlling for other confounding variables that might be correlated with acculturation and/or NL and dietary quality seems a very flawed statistical approach (omitted variables bias). Without controlling for these factors (and even when you control for correlated factors, as it’s very hard to know whether you’ve captured them all, and some may be unobserved/unmeasured), there is no way to know whether it’s, e.g., acculturation that is associated with dietary quality or some other variable that is associated with acculturation and dietary quality that is driving the difference. Claiming that acculturation diminished diet quality and that NL may provide an opportunity for Latina women to either preserve or improve their diet quality, as you did in the abstract, is quite inappropriate in your cross-sectional design and without, at a minimum, controlling for other characteristics/variables that may be associated with acculturation/NL and diet quality using multivariate regression.
Line 105-112: When you are combining acculturation and NL, your sample sizes are really too small to claim differences by group (e.g., n=17 compared to n=8).
Author Response
Response to Reviewer 1 Comments
Point 1: Abstract line 16: Given your cross-sectional study design and statistical approach, I think describing it as “associations between” rather than “differences between” is more accurate.
Response 1: We changed the word differences for associations.
Point 2:Abstract line 17: Please report the sample size, e.g., “This study assessed differences between diet quality, acculturation, and NL of Latina women (n=XXX)…”
Response 2: We added the sample size in the text.
Point 3: Abstract line 18: Please report the p value.
Response 3: P-value reported.
Point 4: Line 32: Seems odd to exclude folate from your list of examples, given its primary role in fetal growth and development.
Response 4: Added folate to the list of nutrients.
Point 5: Line 68: It’s not clear what is meant by “which adds to a global score of 0 to 5.” The acculturation score gets added to what global score?
Response 5: Acculturation global score ranges from 0 to 5, with a maximum global score of 5.
Point 6: Lines 74-75: Annual household income should really be normalized by household size (i.e., an annual household income of 50k for a family of 5 is very different than 50k for a family of two. If you collected household size data, please normalize.
Response 6: We did not collect household size and, unfortunately, cannot normalize the household income.
Point 7: Table 1: Please summarize and test for differences in these characteristics by acculturation and NL groups. My suspicion is that there are statistically significant differences in these characteristics by group, which could be driving your results (see my next comment).
Response 7: We included the distribution by the group as well as the p-value.
Point 8: Lines 103-112: Given your cross-sectional survey design, just testing for statistical differences between groups without controlling for other confounding variables that might be correlated with acculturation and/or NL and dietary quality seems a very flawed statistical approach (omitted variables bias). Without controlling for these factors (and even when you control for correlated factors, as it’s very hard to know whether you’ve captured them all, and some may be unobserved/unmeasured), there is no way to know whether it’s, e.g., acculturation that is associated with dietary quality or some other variable that is associated with acculturation and dietary quality that is driving the difference. Claiming that acculturation diminished diet quality and that NL may provide an opportunity for Latina women to either preserve or improve their diet quality, as you did in the abstract, is quite inappropriate in your cross-sectional design and without, at a minimum, controlling for other characteristics/variables that may be associated with acculturation/NL and diet quality using multivariate regression.
Response 8: Thank you for your suggestions. We included the distribution of the demographic variables in Table 1 and the statistical significance measured by spearman correlations. Based on those analyses, we found that nativity, immigrant status (immigrants vs. non-immigrants), and time living in the US significantly correlated with nutrition literacy and acculturation combined. On table 2, we adjusted the paired two-way ANOVA by immigrant status and time living in the US. We decided not to include nativity since it is a variable derived from nativity with more substantial statistical power. After adjusting the ANOVA model, the associations remained significant, as in lines 103-112.
Point 9: Line 105-112: When you are combining acculturation and NL, your sample sizes are really too small to claim differences by group (e.g., n=17 compared to n=8).
Response 9: We understand that the small sample size and sample in the groups are the limitations of our study. However, considering 1) this is an exploratory study in a population clinically relevant for the impact of acculturation; 2) a group that is recognized as being underrepresented in clinical and population studies (Latina women, mainly Spanish speakers, low acculturation, living in the US more than ten years); 3) the analysis shows a gap in the literature that can lead to further studies with more extensive samples, interventions targeting this underrepresented group, and support policies for healthy eating for Latino women. We consider the analysis relevant to health disparities and minority health science.

Reviewer 2 Report
This is a well-written paper. However, I will suggest authors add a sentence or two to the introduction section to explain why we should care about this research, as outlined in point 1 below. Also, see a few suggestions.
1. What is the take-home here? Why should we care about acculturation and NL and diet quality? Please clarify this in the introduction section.
2. Line 39-40. Can you clarify in which direction the association between NL and the socioeconomic position of pregnant Latino women?
3. Line 82: indicate what was done with outliers if there were any. Was NL and AC normally distributed? If so, indicate it.
4. What is the cut-off for the overall HEI given the total is 100? Will you say the average HEI for this population is adequate?
5. Reference 19 (line 145) was not found on the reference list
Author Response
Response to Reviewer 2 Comments
Point 1: What is the take-home here? Why should we care about acculturation and NL and diet quality? Please clarify this in the introduction section.
Response 1: Thank you for your suggestion. We added the importance of exploring the combined effect of acculturation and NL on diet quality in the introduction lines 43 to 49.
Point 2: Line 39-40. Can you clarify in which direction the association between NL and the socioeconomic position of pregnant Latino women?
Response 2: We clarified the direction of the association between NL and socioeconomic position (higher SEP, higher NL).
Point 3: Line 82: indicate what was done with outliers if there were any. Was NL and AC normally distributed? If so, indicate it.
Response 3: we add the information about normalizing the data and treatment of outliers on lines 91-93
Point 4: What is the cut-off for the overall HEI, given the total is 100? Will you say the average HEI for this population is adequate?
Response 4: HEI scores interpretation is described on lines 68-71. We added a score interpretation and comparison with the average diet of Americans on lines 112-115.
Point 5: Reference 19 (line 145) was not found on the reference list
Response 5: Thank you, we updated the references.

Round 2
Reviewer 1 Report
Unfortunately, I still believe the paper is statistically flawed, and the authors conclusions cannot be supported by their statistical analysis. In particular, their main argument is that their findings show that even though nutrition literacy was not associated with diet quality in the full sample, when disaggregated by acculturation, those with low acculturation and high nutrition literacy had higher dietary quality. My primary concern with this analysis is that the findings have a high likelihood of being spurious due to either (a) the extremely small sample sizes in the compared subgroups (n=17 compared to n=8), and/or (b) the lack of control for confounding variables is this cross-sectional, observational sample (the authors only include nativity and time living in the US). I also find it quite odd/unexpected that the p-values on the subgroup comparison are exactly the same across the three comparisons that are statistically significant (p=.032), though I’m can’t say what this might be attributed to.
Given the very real potential for spurious correlations here rather than actual relationships, I find the authors’ statistical methods and characterization of their findings inappropriate. Also, the authors argument that the small sample sizes should somehow be overlooked “considering 1) this is an exploratory study in a population clinically relevant for the impact of acculturation; 2) a group that is recognized as being underrepresented in clinical and population studies (Latina women, mainly Spanish speakers, low acculturation, living in the US more than ten years); 3) the analysis shows a gap in the literature that can lead to further studies with more extensive samples, interventions targeting this underrepresented group, and support policies for healthy eating for Latino women” doesn’t really make sense to me. Certainly, conducting high-quality research among underrepresented populations is very important and should be prioritized, but conducing low-quality statistical analyses among these underrepresented populations seems unwise, as it can lead to conclusions and policy designs that are unsupported by the realities of these populations.
I think one option would be for the authors to totally re-characterize their analysis as an exploratory, qualitative one and not present statistical differences but rather describe trends, and then, as the authors note in their response to me but not in the paper itself, use those exploratory, qualitative findings to highlight this “gap in the literature that can lead to further studies with more extensive samples, interventions targeting this underrepresented group, and support policies for healthy eating for Latino women.”
Author Response
Response to Reviewer 1 CommentsPoint 1: Unfortunately, I still believe the paper is statistically flawed, and the authors’ conclusions cannot be supported by their statistical analysis. In particular, their main argument is that their findings show that even though nutrition literacy was not associated with diet quality in the full sample when disaggregated by acculturation, those with low acculturation and high nutrition literacy had higher dietary quality. My primary concern with this analysis is that the findings have a high likelihood of being spurious due to either (a) the extremely small sample sizes in the compared subgroups (n=17 compared to n=8), and/or (b) the lack of control for confounding variables is this cross-sectional, observational sample (the authors only include nativity and time living in the US). I also find it quite odd/unexpected that the p-values on the subgroup comparison are exactly the same across the three comparisons that are statistically significant (p=.032), though I can’t say what this might be attributed to.
Response 1: Thank you for your suggestions. We decided to follow your suggestions to re-characterize the analysis as exploratory. We removed any statistical differences and described the data as trends. We also included justification within the text of why the study was exploratory and only descriptive statistics were presented.
Point 2: Given the very real potential for spurious correlations here rather than actual relationships, I find the authors’ statistical methods and characterization of their findings inappropriate. Also, the authors argument that the small sample sizes should somehow be overlooked “considering 1) this is an exploratory study in a population clinically relevant for the impact of acculturation; 2) a group that is recognized as being underrepresented in clinical and population studies (Latina women, mainly Spanish speakers, low acculturation, living in the US more than ten years); 3) the analysis shows a gap in the literature that can lead to further studies with more extensive samples, interventions targeting this underrepresented group, and support policies for healthy eating for Latino women” doesn’t really make sense to me. Certainly, conducting high-quality research among underrepresented populations is very important and should be prioritized, but conducing low-quality statistical analyses among these underrepresented populations seems unwise, as it can lead to conclusions and policy designs that are unsupported by the realities of these populations.
Response 2: Thank you for your comment and suggestions. We agree with you that high-quality research must be conducted with underrepresented populations. We included a language within the manuscript that future studies with larger sample sizes should be developed to explore the impact of the combination of acculturation and nutrition literacy in the diet quality of Latina women during pregnancy.
Point 3: I think one option would be for the authors to totally re-characterize their analysis as an exploratory, qualitative one and not present statistical differences but rather describe trends, and then, as the authors note in their response to me but not in the paper itself, use those exploratory, qualitative findings to highlight this “gap in the literature that can lead to further studies with more extensive samples, interventions targeting this underrepresented group, and support policies for healthy eating for Latino women.”
Response 3: Thank you for your suggestion. We re-characterized our analysis for exploratory and reported the data and results as trends instead of statistically significant. We also added within the text the information about the gap in the literature.
